# On-Chip Cell Incubator for Simultaneous Observation of Culture with and without Periodic Hydrostatic Pressure

**DOI:** 10.3390/mi10020133

**Published:** 2019-02-17

**Authors:** Mitsuhiro Horade, Chia-Hung Dylan Tsai, Makoto Kaneko

**Affiliations:** 1Department of Mechanical Engineering, Osaka University, Suita 565-0871, Japan; mk@mech.eng.osaka-u.ac.jp; 2Department of Mechanical Engineering, National Chiao Tung University, Hsinchu 30010, Taiwan; dylantsai@nctu.edu.tw

**Keywords:** on-chip cell incubator, periodic hydrostatic pressure, periodic pressure, time-lapse observation, cell growth, simultaneous multiple chamber observation

## Abstract

This paper proposes a microfluidic device which can perform simultaneous observation on cell growth with and without applying periodic hydrostatic pressure (Yokoyama et al. *Sci. Rep.*
**2017**, *7*, 427). The device is called on-chip cell incubator. It is known that culture with periodic hydrostatic pressure benefits the elasticity of a cultured cell sheet based on the results in previous studies, but how the cells respond to such a stimulus during the culture is not yet clear. In this work, we focused on cell behavior under periodic hydrostatic pressure from the moment of cell seeding. The key advantage of the proposed device is that we can compare the results with and without periodic hydrostatic pressure while all other conditions were kept the same. According to the results, we found that cell sizes under periodic hydrostatic pressure increase faster than those under atmospheric pressure, and furthermore, a frequency-dependent fluctuation of cell size was found using Fourier analysis.

## 1. Introduction

There are many studies addressing cellular responses under mechanical and chemical stimulations [1,2,3,4,5,6,7]. Stimulation can be beneficial for cultured cells. For example, Di Carla et al. used caffeine as a xenobiotic stress-inducing agent and found the cell survival rate is promoted under such a chemical stress [8]. Some works used stimulations as a tool for determining cellular properties or physiologies. For example, Seo et al. showed the relationship between mechanical stimulation and the physiologies in a dystrophic heart [9]. Ito et al. applied mechanical stress to red blood cells (RBCs) for different durations and observed a 100 times difference of the time constant in their shape-recovery curves when the stress duration was just increased from 10 s to 180 s [7]. Sakuma et al. proposed a cell stress test by moving a RBC back and forth across a microfluidic constriction until the RBC eventually lost its deformability and used it as an index of RBC deformability [10]. There are also studies investigating cell alignment under stimulations. For example, Teramura et al. demonstrated that mechanical stimulation to human iPS cells altered the alignment of actin fibers as well as the expressions of the pluripotent related genes [11]. Subramony et al. investigated the role of nanofiber matrix alignment and mechanical stimulation on mesenchymal stem cell (MSC) differentiation [12]. While there are approaches using 3D printing for aligning cells [13,14,15], stimulation seems to be a more natural approach, since each cell can choose its comfortable position and orientation with fewer constraints.

Culture with periodic hydrostatic pressure here refers to a periodic stimulation of mechanical stress onto cells during cell culture. The concept of such a periodic hydrostatic pressure is can be explained with the analogy to the movement during a human workout. An interesting result was recently presented by Yokoyama group, who utilized periodic pressure during culturing smooth muscle (SMC) cells. They found an optimum recipe regarding culturing elastic cell sheet, and the applied frequency, the minimum pressure, and the maximum pressure of the recipe were 0.002 Hz, 110 kPa, and 180 kPa, respectively [16]. Under such an optimum recipe, they succeeded in growing a 10-layer cell sheet in 20 days. Furthermore, they made a vascular graft, with a diameter of 1 mm, from the cell sheet and transplanted the graft into a rat for reconnecting a cut artery. The results showed that the rat could continuously survive for 2.5 months after the transplantation, and new capillaries were found grown on the transplanted tissue. This is a sign showing that the body can successfully adapt to transplanted tissue and apply the necessary nutrition to the part.

The first prototype of on-chip cell incubator was recently presented with a result showing that the cells cultured with periodic hydrostatic pressure have a greater number of stress fibers [17]. While cell sheet cultured under periodic hydrostatic pressure is having a greater elasticity than that under atmospheric pressure [16], it is still not clear when and how this difference happens. The main goal of this work is to reveal the difference between cultured cells with and without periodic hydrostatic pressure during the culture. We focused on the cell behavior in the first one hour after cell seeding. The cell behavior was observed with time-lapse images taken from the observation window of a developed on-microscope incubation system. The proposed cell incubator was with two independent culture arrays in it. The key advantage is that we can compare both cell responses under the same conditions except periodic hydrostatic pressure. Through the experiments, we found that cells showed remarkable size growth under periodic hydrostatic pressure with the frequency-dependent fluctuation that the cell size changed with the applied pressure.

## 2. Materials and Method

### 2.1. Periodic Hydrostatic Pressure and Optimum Stimulus

Figure 1a illustrates the analogy of human exercise to cell culture with periodic hydrostatic pressure. The upper and lower figures in Figure 1a are examples of human exercise with weight lifting and cell culture with periodic hydrostatic pressure, respectively. Figure 1b,c explains how cells are cultured under atmospheric pressure and under periodic hydrostatic pressure. The rightmost pictures of Figure 1b,c are our preliminary results of cultured cells. In the results, stress fibers, the actin filaments, were stained with Acti-stain 488 Fluorescent phalloidin for the convenience of observation. It can be seen that the stress fibers of the cells under periodic hydrostatic pressure are thicker than those under atmospheric pressure. The results in Figure 1b,c show clear difference with and without periodic hydrostatic pressure.

The three most critical parameters of periodic hydrostatic pressure, are the maximum pressure, the minimum pressure, and frequency. These three parameters control the pressure pattern with respect to time during cell culture. In conventional works of pressure-based stimuli culture, the frequencies of periodic pressure are mostly around 1 Hz, just like heartbeat and breathing [16,18,19,20,21]. Different from those studies, we tested low frequency zone much less than 1 Hz and reached a new optimum frequency, which had not yet been found previously. As for the determination of the pressure pattern, we had to determine both the maximum and minimum pressures in addition to the frequency. It is well known that our blood pressure is slightly higher than atmospheric pressure, it is about 100 kPa. When the minimum value of blood pressure, which is also known as diastolic blood pressure, is roughly 80 mmHg, about 10 kPa. Thus, we set up the lower pressure is 110 kPa which corresponds to human minimum blood pressure. As for the maximum pressure, we selected it based on the gene expression, such as Fiburin and Lysyl [16]. Fiburin and Lysyl are two important components for stress fibers of human smooth muscle cells (HSMCs), where Fiburin is a key component of elastic fiber growth, and Lysyl helps cross-linking. After preliminary experiments, the maximum pressure of 180 kPa, roughly equivalent to 600 mmHg and about five times the blood pressure, was chosen [16]. Overall, the parameters of the maximum pressure, minimum pressure, and frequency were 180 kPa, 110 kPa, and 0.002 Hz, respectively. The time period of one cycle is about 8 min and 20 s. This means that the periodic pressure between 180 kPa and 110 kPa is given for every 250 s. Of course, we would like to make frequency lower than this, but it could not be done because the pH balance is broken due to pressurized CO_2_. If pressurized continues for long time, CO_2_ continues to dissolve into the culture solution, and pH becomes below 5.8, while the optimal cell culture pH is between 5.8 and 6.2.

### 2.2. Development of Cell Incubator and Pressure Control

We aimed to compare cell behaviors with and without periodic hydrostatic pressure during cell culture in this work. To do this, an important issue was how to simultaneously observe the cell groups. We propose the on-chip cell incubator where we can observe what happens in real-time on a single chip, as shown in Figure 2. The chamber arrays on the chip are placed in symmetry to the center line, and they are connected to two independent microfluidic channels. Through this device, we can impart pressure on cells with specified pressure patterns. Microfluidic devices have often been used for experiments under a microscope [22,23,24]. For example, Eyer et al. used a microchamber array for single cell isolation and analysis of intracellular biomolecules [25]. Reaction experiments and local irritation experiments on single cells can be performed using a microfluidic chip. The on-chip cell incubator in this method was fabricated using a standard photolithography approach with microelectromechanical systems (MEMS) technology and is made of polydimethylsiloxane (PDMS). The fabrications details of the PDMS chip can be found in Appendix B.

The system is composed of a microscope (OLYMPUS: IX71, Olympus Co., Tokyo, Japan), a commercial incubator (SCA-80DS, ASTEC Inc., Fukuoka, Japan), a compressor (DOP-80SP, ULVAC Inc., Kanagawa, Japan), a flow regulator (ITV2030-212BL, SMC Co., Tokyo, Japan), a pressure sensor (HP101-C31-L50A*B/V1, Yokogawa Electric Co., Tokyo, Japan), a digital CMOS camera (C11440, Hamamatsu Photonics K.K., Shizuoka, Japan), and an on-microscope cell incubation system. The on-microscope incubation system is used for maintaining a culture environment as in the leftmost photo in Figure 2. The temperature around the chip is controlled by a feedback system using a heater and a temperature sensor to maintain an environment of 37 °C. The concentration of CO_2_ is controlled at 5% by drawn gas from the commercial incubator to the on-microscope incubation system by the flow regulator. Both the inlet and outlet of the chip are connected with Teflon tubes to the pressure regulator, so that the inside of the micro chambers can be pressurized with a specified level at specified time sequences. As mentioned in the previous section, a pressure of 180 kPa and 110 kPa was applied as the periodic hydrostatic pressure in this work.

### 2.3. Chip Preparation and Experimental Procedure

In order to adhere cells to the PDMS surface for cell culture, 10% adhesion-assist protein fibronectin was coated on the surface of culture array by injecting 25 µL of it from the inlet of the device using a micropipette. The area inside of the PDMS flow channel and micro chamber arrays was filled with the solution and placed at 37 °C in the incubator for 60 min. Afterwards, the fibronectin solution was removed and rinsed with phosphate buffered salts (PBS).

Human smooth muscle cells (HSMCs) were used for all experiments in this work. Cells were injected into the chamber arrays for seeding from the inlets of the PDMS device. The concentration of HSMC was 2.5 × 10^5^ cells/mL. Low cell density was used for the convenience of single cell analysis, particularly for avoiding overlaps of the projected area. The tube on the outlet side of the periodic hydrostatic pressure chamber was connected to the compressor for pressurized gas. Since the height of the micro chamber is higher than the other flow micro channel region, more cells can be trapped in the micro chamber due to inertial flow. After cell injection, the cells were gradually attached to the bottom of the glass by gravity force. For the culture with periodic hydrostatic pressure, the tube on the inlet side was sealed and periodic pressure was applied to the chamber. For the culture without periodic hydrostatic pressure, the tube was open to the atmosphere.

Time-lapse imaging was employed for comparing the difference between cell growth with and without periodic hydrostatic pressure. The camera takes a picture covering both chambers simultaneously every 10 s after seeding. We focused on how the cells grow during the first hour after seeding. Figure 3a shows an example of a captured time-lapse image. Both cells in the two micro chambers were recorded simultaneously, where the upper and lower chamber in Figure 3 are with and without periodic hydrostatic pressure mode, respectively. Only about 5–10 cells were trapped in the chamber of the on-chip cell incubator device for the convenience of single cell observation.

In order to evaluate the degree of cell growth, we introduced the projected area as an evaluation index where it was defined by the projected area of cell in the horizontal plane. An example of cell growth and image processing is demonstrated in Figure 3b,c, respectively. In Figure 3b, cell images were acquired for each record time, and images at 20 min, 40 min, and 60 min. Figure 3c shows the image at 60 min after seeding and the left shows the original image. The outline in the middle of Figure 3c allows us to determine the cell area, as shown in the rightmost image in Figure 3c. As a result, we can obtain the information of cell shape and area using image analysis software *Image J* (1. 50i, Wayne Rasband, National Institutes of Health, USA).

Figure 4 shows measured results of pressure control during the culture, and the pressure was cycling between 180 kPa and 110 kPa every 250 s. Figure 4a,b is close views of increasing and decreasing pressure periods, respectively. From Figure 4a,b, we can see a reasonably well controlled pressure where an error is roughly less than 1% with respect to the target value without overshoot. In addition, it can be seen that the time for pressurization to the target values is less than one second.

## 3. Results

### 3.1. Projected Area of Cells with and without Periodic Hydrostatic Pressure

Figure 5 shows the growth of the projected cell area with respect to time where Figure 5a,b denotes the projected areas and the average value among the six trapped cells, respectively. The cell groups with and without periodic hydrostatic pressure are indicated by red and blue marks and the original of the time axis is the starting time of pressurization. From Figure 5, no significant difference can be seen between the two groups with and without periodic hydrostatic pressure.

An example of cell area changes under periodic hydrostatic pressure is shown in Figure 6, where there is a remarkable point (b). The projected cell area increased rapidly from (b) to (c), and was with nearly three times faster than the initial 15 min. After that, the cell entered another phase where the projected cell area increased with a slightly gentle slope from (c) to (d). It is interesting to know the tendency of the projected cell area with or without periodic hydrostatic pressure with such instances. A time-lapse cell behavior during cell culture can be found in the Appendix A. Appendix A shows the cell behavior from the point (a) to the point (d), and Appendix A shows the cell behavior after the point (c). After point (c), where rapid growth ended, interesting behavior was observed in which the cell periodically extends in terms of its size.

For determining point (b) and point (c), we used S(*t*), defined by:(1)$$S\left(t\right)=\frac{A_{\left(t\right)}-A_{min}}{A_{max}-A_{min}}$$
where A*_min_*, A*_max_*, and A(*t*) are the projected cell areas at the time in point (b), at the time in point (c), and at the time of t, respectively. Simply speaking, point (b) is the starting point when the cell size starts to increase rapidly and point (c) is the ending point when the rapid change of cell size is terminated.

Figure 7 shows the normalized S(*t*), where the cell groups with and without periodic hydrostatic pressure are indicated by red and blue marks, respectively. The origin of the horizontal axis is the time which corresponds to the point (b) in Figure 6. The velocity of the projected cell area increased dramatically after point (b). From Figure 7, both cell groups with and without periodic hydrostatic pressure resulted in 1.0 in about 20 min (1200 s), which matched well with the definition of A*_min_* and A*_max_*. On the other hand, the tendencies of S(*t*) shown by the two groups after 20 min (1200 s) were different. The cell group with periodic hydrostatic pressure continued to increase the projected cell area with a gentle slope, whereas the cell group without periodic hydrostatic pressure was only a tiny positive slope. The *p*-value of the T test for the cell size of two groups at 20 min (1200 s) was 0.173 and indicated no significant difference. However, when the time reached 30 min (1800 s), the *p*-value became less than 0.05 and demonstrated a significant difference between the growth rate with and without periodic hydrostatic pressure. In other words, the significant difference for the culture with and without periodic hydrostatic pressure happened after the time past 30 min in Figure 7, as the point (c) in Figure 6.

### 3.2. Periodic Characteristics in Cell Growth

Figure 7 shows two important results, one of which is that there is statistically meaningful difference at 30 min after cell extension, and the other is that the growth pattern is fluctuating with respect to time. To observe the growth pattern more qualitatively, let us rearrange the time domain, so that we can adjust the phase among all experiments under periodic hydrostatic pressure.

Figure 8 shows the normalized projected cell area S(*t*), where the origin of the horizontal axis is the time corresponding to (c) in Figure 6, more specifically when the pressure is switched from 180 kPa to 110 kPa. This means that A*_max_* in Equation (1) is the projected cell area at the time t when the pressure is switched from 180 kPa to 110 kPa in the nearest time around (c) in Figure 6.

### 3.3. Frequency Analysis on Projected Area of Cells

Figure 9 explains how to achieve the frequency analysis for one particular cell under periodic hydrostatic pressure, where Figure 9a–c denotes the normalized area with respect to time, the curve defined by ΔS(*t*) = S(*t*) − S(*t*)_approximate curve_, and the frequency analysis, respectively. S(*t*)_approximate curve_ is obtained by a linear fit with the-least-squares method as shown in Figure 9a. From Figure 9c, we can see an interesting observation, namely, frequency-dependent cell growth. The frequency of 0.002 Hz was a peak of the frequency analysis and it corresponds to the frequency of periodic hydrostatic pressure. Figure 9c–h shows three examples of frequency analysis where (c) through (e) is cultured under periodic hydrostatic pressure and (f) through (h) is cultured under atmospheric pressure. In the cell group cultured under periodic hydrostatic pressure, we can see clear amplitude in the frequency domain with 0.002 Hz, while we cannot see clear amplitude in the frequency domain with 0.002 Hz in the cell group cultured under atmospheric pressure.

## 4. Discussion

The on-chip cell incubator proposed in this paper includes two important key words, one of which is “same tme history” and the other is “simultaneous observation”. “Same time history” means cells whose initial conditions are exactly the same, including the time for all cells in two chambers. It is a great advantage to completely avoid the influence for the results coming from the time difference among cells, and therefore, we can keep the culture condition the same in both chambers except either under periodic hydrostatic pressure or atmospheric pressure. “Simultaneous observation” allows us to visualize cell behaviors coming from only one parameter, in this work the effect of periodic hydrostatic pressure.

Using the on-chip cell incubator, we could observe how the cells grow by focusing in the first hour for both cultures with and without periodic hydrostatic pressure. The most interesting result under periodic hydrostatic pressure is that cells in growth period increase the projected cell area according to the pressure frequency imparted on the culture liquid. This effect is more enhanced for the cell whose size is bigger. To the best of the authors’ knowledge, there was no such size fluctuation of cell size reported in literature under a periodic pressure stimulus. A natural question that comes up is why there has been no report on the frequency-dependent cell size fluctuation so far. Our work is based on an extremely low frequency, 0.002 Hz, while the frequencies of periodic pressure in former works are mostly around 1Hz, just like heartbeat and breathing [16,18,19,20,21]. We believe that several minutes are needed for cells to change size during periodic hydrostatic pressure and it is hard to change the size with a noticeable range under a pressure frequency with around 1 Hz.

## 5. Conclusions

In the same way that human muscle grows after exercise, it is known that an elastic cell sheet can be obtained by cell culture with periodic hydrostatic pressure. This paper presented simultaneous cell observation by an on-chip cell incubator with and without such a periodic hydrostatic pressure. The periodic pressure with an extremely low frequency of 0.002 Hz was imparted for one chamber and atmospheric pressure was given for the other one. The experiments were only focused on the first one hour after cell seeding, and significant difference of cell growth were observed. We also found an interesting phenomenon during periodic hydrostatic pressure where the projected areas of the cells increased at the same frequency as the pressure frequency imparted on them. For future work, we plan to test with difference frequencies and see in which frequency the frequency-dependent cell growth disappears.

## Figures and Tables

**Figure 1 micromachines-10-00133-f001:**
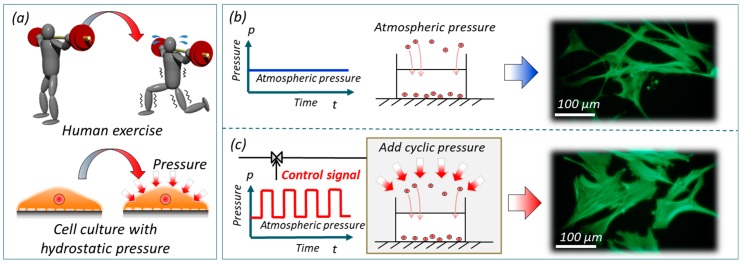
An overview of Periodic hydrostatic pressure. (**a**) The analogy of Periodic hydrostatic pressure to human exercise. (**b**) Conventional cell culture and a result showing the stress fibers of the cultured human smooth muscle cells (HSMCs) under atmospheric pressure. (**c**) Cell culture with Periodic hydrostatic pressure and a result showing the stress fibers of the cultured HSMCs under periodic pressure. The stress fibers, the actin filaments, were stained with Acti-stain 488 Fluorescent phalloidin.

**Figure 2 micromachines-10-00133-f002:**
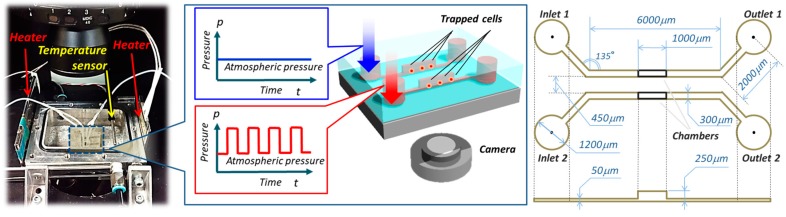
An overview of the experimental system. From left to right are the on-microscope incubation system, design of on-chip cell incubator and its dimensions, respectively. The system is mainly composed of two parallel chambers for simultaneously observation of cell culture with and without Periodic hydrostatic pressure.

**Figure 3 micromachines-10-00133-f003:**
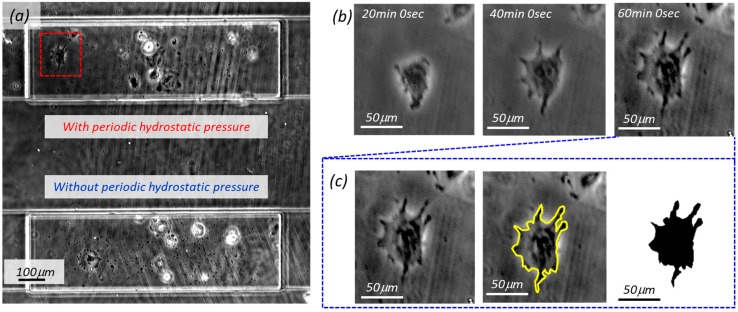
Captured pictures by time-lapse imaging. (**a**) Captured microscopic image where the chambers on the top and the bottom are the cells cultured with and without the proposed periodic hydrostatic pressure, respectively. An example of cell assessment is demonstrated using the highlighted cell. (**b**) Selected time-lapse images of the cell at the time of 20, 40, and 60 min. (**c**) The procedure to obtain the extracted cell area.

**Figure 4 micromachines-10-00133-f004:**
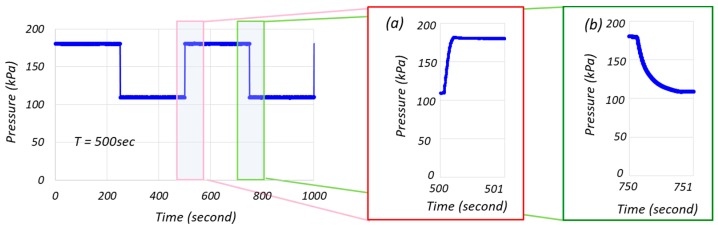
The performance of pressure control during periodic hydrostatic pressure. (**a**) A close view of the pressure increasing period. (**b**) A close view of the pressure decreasing period.

**Figure 5 micromachines-10-00133-f005:**
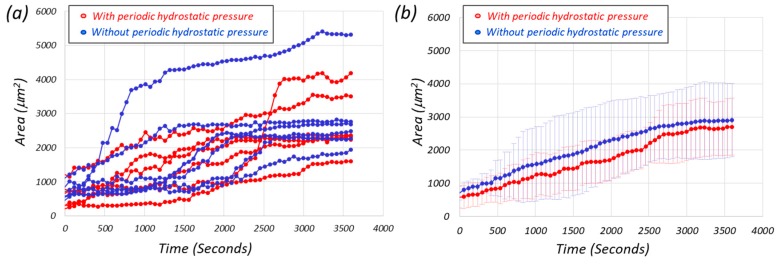
The growth of the projected area with respect to time, where the cell groups with and without periodic hydrostatic pressure are indicated by red and blue marks, respectively. (**a**) Measured area of six cells from the periodic hydrostatic pressure chamber (red) and 6 cells from control (blue), a culture chamber without periodic hydrostatic pressure (**b**) The average value and standard deviation from the six cells in each chamber are plotted. No significant difference between two.

**Figure 6 micromachines-10-00133-f006:**
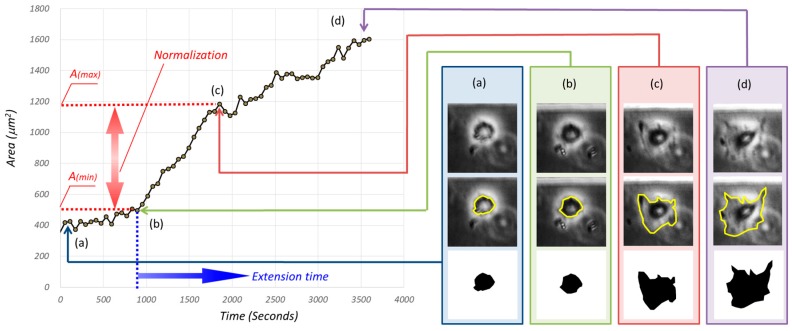
An example of the projected cell area with respect to time. (**a**,**b**) are the cell images at different instance, and from the top to the bottom are the original cell image, contour extraction, the projected cell area, respectively. It should be noted that the increase velocity of the projected cell area from (b) to (**c**) is larger than that of other phases, such as from (c) to (**d**).

**Figure 7 micromachines-10-00133-f007:**
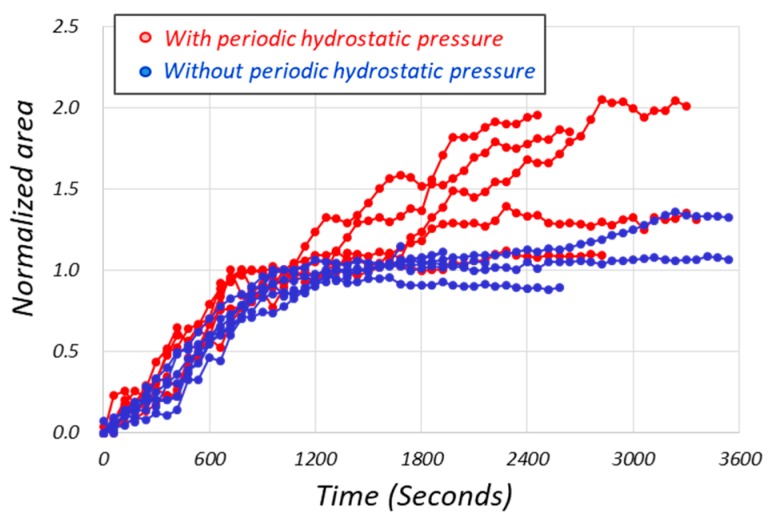
The normalized projected cell area with respect to the time where the origin of the time is redefined by the time corresponding to point (b) in Figure 6. The cell groups with and without periodic hydrostatic pressure are indicated by red and blue marks, respectively, and normalized for each of the six original data shown in Figure 5.

**Figure 8 micromachines-10-00133-f008:**
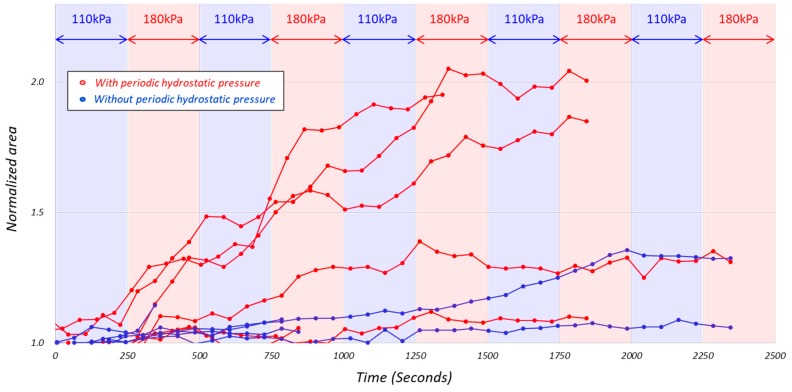
The normalized projected cell area S(*t*) where the origin of the horizontal axis is the time corresponding to (c) in Figure 6, more specifically when the pressure is switched from 180 kPa to 110 kPa in the nearest time around (c) in Figure 6, and normalized for each of the six original data shown in Figure 5.

**Figure 9 micromachines-10-00133-f009:**
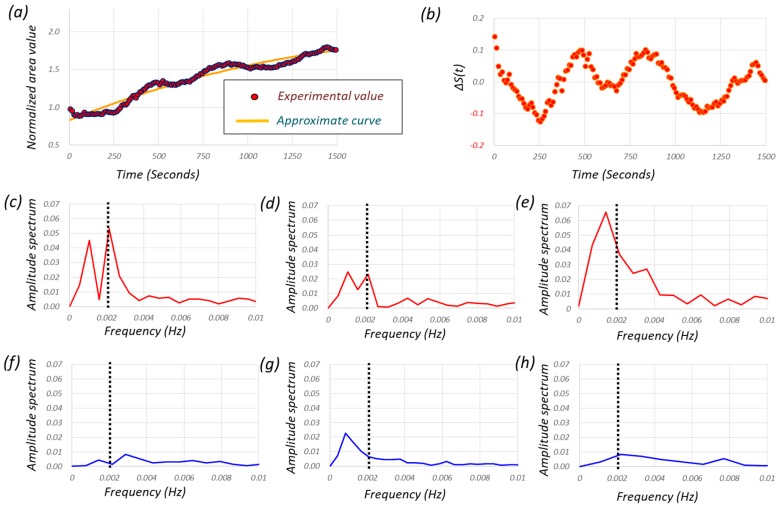
How to achieve frequency analysis, where (**a**) is the normalized area with respect to time where the approximate curve is shown by the line shown by red color, (**b**) the curve defined by S(*t*) − S(*t*)_approximate curve_, and (**c**) the frequency analysis. The frequency analysis, where (**c**–**e**) are under periodic hydrostatic pressure, and (**f**–**h**) are cultured under atmospheric pressure. Three of each of the six original data were carried out.

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
