# Peer review of "On-Chip Cell Incubator for Simultaneous Observation of Culture with and without Periodic Hydrostatic Pressure"

_micromachines, 2019, doi:10.3390/mi10020133_

Round 1

Reviewer 1 Report

This study describes the development of a device on which cellular morphology under a normal and pressurized condition can be monitored. The authors called this device as “On-Chip Cell Gym” since the pressurized environment can facilitate the spreading of cells, especially human smooth muscle cells in this study. Overall, this manuscript is well organized and written. Furthermore, the process of the experiment including the development of the device, the set-up of the experimental conditions, and the systemic analysis of cell morphologies. The writing is also clear and the implication of the experimental results is also well described. Therefore, I believe this manuscript can be published in Micromachines Journal without further modification. 

Author Response

Dear Editor and Reviewers,

              We would like to thank you for the timely review as well as constructive comments to our manuscript. We have greatly revised the manuscript based on the reviewers’ comments. The main modifications include

1.      The abstract and introduction are completely re-organized and re-written.

2.      The structure of the work is re-organized in a conventional format.

3.      Figures are removed and combined accordingly.

4.      Two supplementary video of experimental results are attached for the readers.

5.      The typos are corrected and English is polished.

The point-by-point responses to all the comments are as follows:

Reviewer #1

This   study describes the development of a device on which cellular morphology   under a normal and pressurized condition can be monitored. The authors called   this device as “On-Chip Cell Gym” since the pressurized environment can   facilitate the spreading of cells, especially human smooth muscle cells in   this study. Overall, this manuscript is well organized and written.   Furthermore, the process of the experiment including the development of the   device, the set-up of the experimental conditions, and the systemic analysis   of cell morphologies. The writing is also clear and the implication of the   experimental results is also well described. Therefore, I believe this   manuscript can be published in Micromachines Journal without further   modification.

Response   1-1:

Thanks   to the reviewer for the positive comments. We really appreciate it.

Reviewer 2 Report

This manuscript presented the development of a microfluidic device which contained two culture chambers with different liquid pressure. By using this system, the authors investigated the spread behavior of smooth muscle cells with and without “Cell Exercise”. It is an interesting topic to study the effect of mechanical stress on cell behavior. However, the manuscript suffers from the lack of a good organized form. The organization of both text description and figures is not an appreciate method. Furthermore, the data presentation of this manuscript is preliminary, which does not provide enough information to show the advantages of this method. Therefore, I recommend it needs major revisions. The specific comments are listed as below.

1.     The form of the paper needs to be well organized in an appreciate way. Instead of 8 sections in this manuscript, the normal organized form of the paper is “Introduction, Materials and Methods, Results, Discussions, and Conclusions”. The readers could read the paper and find their interested points easily if the paper followed this form.

2.     The “Introduction” part needs to be re-written. The authors should summary the relative works and present the normal approaches, results and requirements of this research area, rather than simplify listed the works one by one.

3.     The figures should be organized well too. 11 figures are too many for a research article. The authors need to compact the results into 6-7 figures with more information in each one. For example, figures 1-3 are all schematic diagrams, which can be compacted into one figure. And in figure 8, there no necessary to present the same results as two diagrams of individual data and average data.

4.     The figure legends were too brief. At least, the essential information such as the number of repeat samples (N=?), the data presented mothed (Mean±S.D. or S.E?), sample treatment time et al. should be presented in figure legends.

5.     In figure 4, it seems only one cell spread in each chamber and the cell density was very low. Could the authors supply more results about the cell viability in the chip chamber?

6.     Why could the pressure frequency affect the cell spread behavior? Will the pressure frequency influence other behaviors of cell? The authors should talk more about the mechanism in the discussion part.

7.     When the authors doing Time-Lapse Imaging, did a live cell working station be used? If not, how did the authors maintain the temperature of the media in the chip? 

8.     The whole manuscript needs to be checked carefully. There were some subscripts and symbols displayed in a wrong way, such as “μm”. Furthermore, the language needs to be improved too.

Author Response

Dear Editor and Reviewers,

              We would like to thank you for the timely review as well as constructive comments to our manuscript. We have greatly revised the manuscript based on the reviewers’ comments. The main modifications include

1.      The abstract and introduction are completely re-organized and re-written.

2.      The structure of the work is re-organized in a conventional format.

3.      Figures are removed and combined accordingly.

4.      Two supplementary video of experimental results are attached for the readers.

5.      The typos are corrected and English is polished.

The point-by-point responses to all the comments are as follows:

This manuscript   presented the development of a microfluidic device which contained two   culture chambers with different liquid pressure. By using this system, the   authors investigated the spread behavior of smooth muscle cells with and   without “Cell Exercise”. It is an interesting topic to study the effect of   mechanical stress on cell behavior. However, the manuscript suffers from the   lack of a good organized form. The organization of both text description and   figures is not an appreciate method. Furthermore, the data presentation of   this manuscript is preliminary, which does not provide enough information to   show the advantages of this method. Therefore, I recommend it needs major   revisions. The specific comments are listed as below.

1. The form of the   paper needs to be well organized in an appreciate way. Instead of 8 sections   in this manuscript, the normal organized form of the paper is “Introduction,   Materials and Methods, Results, Discussions, and Conclusions”. The readers   could read the paper and find their interested points easily if the paper   followed this form.

Response 1

Thank you for the   suggestions. The manuscript has been re-organized accordingly.

2. The “Introduction”   part needs to be re-written. The authors should summary the relative works   and present the normal approaches, results and requirements of this research   area, rather than simplify listed the works one by one.

Response 2

The introduction part   has been re-organized and re-written. Descriptions of related research and   this research are added.

3. The figures should   be organized well too. 11 figures are too many for a research article. The   authors need to compact the results into 6-7 figures with more information in   each one. For example, figures 1-3 are all schematic diagrams, which can be   compacted into one figure. And in figure 8, there no necessary to present the   same results as two diagrams of individual data and average data.

Response 3

Figure 3 is removed as   it can be explained with Figure 2. Also, Figure 8(b) is deleted as   suggested. We also combined Figures 10 and 11 for the convenience of reading.   Since Figures 1 and 2 show the conceptual diagram and the experimental   apparatus, respectively, we decided to keep them.

4. The figure legends   were too brief. At least, the essential information such as the number of   repeat samples (N=?), the data presented mothed (Mean±S.D. or S.E?), sample   treatment time et al. should be presented in figure legends.

Response 4

Descriptions of the   figure have been enhanced for improving the readability.

5. In figure 4, it   seems only one cell spread in each chamber and the cell density was very low.   Could the authors supply more results about the cell viability in the chip   chamber?

Response 5

The low cell density   is tested for the convenience of single cell analysis, particularly for   avoiding overlaps of projected area. The description is added in Line   131-135.

The continuous growth   of cell, such as the projected area shown in Fig.3(b), support the viability   of the cell. In addition, the comparison in Fig.7 shows similar viability of   cells with and without Cell Exercise, and that indicates the similar growth   status between the experimental group and control group.

6. Why could the   pressure frequency affect the cell spread behavior? Will the pressure   frequency influence other behaviors of cell? The authors should talk more   about the mechanism in the discussion part.

Response 6

The mechanism of why   the pressure affecting the cell behavior is not yet clear and is one of the   reasons for developing the proposed Cell Gym, so that we can directly observe   the difference with and without Cell Exercise. According to our previous   work, we have found that Cell Exercise enhanced gene expression on FN,   Fibrillin-1, Fibrillin-2, Fibulin-4 and Lysyl oxidase [1]. In the future we   will find more details by establishing a method to directly observe   dephosphorylation and cytoskeleton. The above content was added to the   discussion.

7. When the authors   doing Time-Lapse Imaging, did a live cell working station be used? If not,   how did the authors maintain the temperature of the media in the chip?

Response 7

Thank you for the   comment. The information of the on-microscope incubation system was missing   in the submitted manuscript. We use feedback system with heater and   temperature sensor to maintain environment of 37 ℃ as well as the 5% CO2   concentration around chip. It is added in Figure 2 and the text in Sec. 2.2.

8. The whole   manuscript needs to be checked carefully. There were some subscripts and   symbols displayed in a wrong way, such as “μm”. Furthermore, the language   needs to be improved too.

Response 8

Our apology for the   careless mistakes. We have carefully checked the subscripts, symbols and polished   the language accordingly.

Reviewer 3 Report

The manuscript has numerous flaws.

In brief:

- English must be revised and edited by a native speaker.

- The overall structure of the manuscript is unconventional which makes it hard to follow. It is also hard to distinguish between what has been done in this manuscript from what has been presented in previous articles... for example, it is not clear if the results presented in the figure 1 are from a previous article or from this manuscript.

- Some statements are overstated and pompous (e.g.: "We believe that this is the first experimental evidence in the research history of Cell Exercise."… the “history” of “Cell Exercise” is too limited… 5 papers in the reference list, none of them is a review article, and none of them is mentioning pressure as a mean for “cell exercise”).

- The authors should provide a better explanation (with appropriate references) of why they chose the points (b) and (c) for their analysis. Is it based on others' work? Is it a conventional way? There is surely a scientific explanation, but it is missing in the manuscript. Unfortunately, as it is, it misguides the reader in thinking that the authors performed a first round of results analysis and did not find differences, so, they dug a little bit and picked points that allow them to support their idea with significant differences.

- Preliminary data described in the manuscript are missing (not found in the supplementary information neither).

- The authors present applying pressure to the cells as a cell exercise similar to human exercise. However, "human" exercise relies on cell contraction and extension (as shown in their figure 1) not on pressure. An analogy to their system would be to apply pressure to a lied-down human body and see if the overall "projected" area will increase (it surely will, with enough pressure). I would not call that "exercise", though, but “flattening”.

In addition, “exercises” are supposed to make humans healthier. The authors did not demonstrate that applying cyclic pressure increases cell “health” (“better” viability, “better” cell marker expression, etc …), which would support their idea of “cell exercise”.

- The scientific goal is unclear. What is the purpose of the manuscript? Introducing a new chip?

Author Response

Dear Editor and Reviewers,

              We would like to thank you for the timely review as well as constructive comments to our manuscript. We have greatly revised the manuscript based on the reviewers’ comments. The main modifications include

1.      The abstract and introduction are completely re-organized and re-written.

2.      The structure of the work is re-organized in a conventional format.

3.      Figures are removed and combined accordingly.

4.      Two supplementary video of experimental results are attached for the readers.

5.      The typos are corrected and English is polished.

The point-by-point responses to all the comments are as follows:

The manuscript has   numerous flaws.

In brief:

- English must be revised   and edited by a native speaker.

Response 1

Our apology for the typos   and writing. We have carefully checked the subscripts, symbols and polished   the language accordingly.

- The overall   structure of the manuscript is unconventional which makes it hard to follow.   It is also hard to distinguish between what has been done in this manuscript   from what has been presented in previous articles... for example, it is not   clear if the results presented in the figure 1 are from a previous article or   from this manuscript.

Response 2

Thank you for your   comment. We have completely re-structure the manuscript to a conventional   arrangement. Figure 1 is first shown in this manuscript and the fluorescent   images are a repeating test based on our previous results in [1].  Description regarding the fluorescent images   in Fig.1 is added in introduction.

- Some statements are   overstated and pompous (e.g.: "We believe that this is the first   experimental evidence in the research history of Cell Exercise."… the   “history” of “Cell Exercise” is too limited… 5 papers in the reference list,   none of them is a review article, and none of them is mentioning pressure as   a mean for “cell exercise”).

Response 3

Thank you for your   comment. We have revised the description for avoiding misunderstanding,   especially in the items of discussion and conclusion. The concept of cell   exercise appeared in our previous conference proceedings (MEMS2017) and the   proceeding is cited in this work.

- The authors should   provide a better explanation (with appropriate references) of why they chose   the points (b) and (c) for their analysis. Is it based on others' work? Is it   a conventional way? There is surely a scientific explanation, but it is   missing in the manuscript. Unfortunately, as it is, it misguides the reader   in thinking that the authors performed a first round of results analysis and   did not find differences, so, they dug a little bit and picked points that   allow them to support their idea with significant differences.

Response 4

Thank you for your   important comments. Factors focusing on points (b) and (c) are based on   experimental observation. We found that all cells had similar growing   patterns once it started to expand on the substrate, and as a result, we set   (b) as a new origin of time for all cells and unexpectedly found the   consistent pattern as shown in Fig.7. It is actually reasonable because   individual cells may settle on the substrate at different instance, so the   starting point of cell attachment may different from cell to cell. Based on   the results, originating all growing curve at (b) seems to be fair. The   descriptions are added to the section of analysis.

- Preliminary data   described in the manuscript are missing (not found in the supplementary   information neither).

Response 5

As preliminary data,   movies and descriptions added to the above items are added.

- The authors present   applying pressure to the cells as a cell exercise similar to human exercise.   However, "human" exercise relies on cell contraction and extension   (as shown in their figure 1) not on pressure. An analogy to their system   would be to apply pressure to a lied-down human body and see if the overall   "projected" area will increase (it surely will, with enough   pressure). I would not call that "exercise", though, but   “flattening”. In addition, “exercises” are supposed to make humans healthier.   The authors did not demonstrate that applying cyclic pressure increases cell   “health” (“better” viability, “better” cell marker expression, etc …), which   would support their idea of “cell exercise”.

Response 6

The   “exercise” here is not defined in a strict manner as the reviewers pointed   out, but rather an easy-to-understand concept that mechanical stimulus   strengthen cells as its analogy to human exercise. According to our previous   work in [1], such a periodic pressure effectively improve the elasticity of   cell sheet.

- The scientific goal   is unclear. What is the purpose of the manuscript? Introducing a new chip?

Response 7

The primary objective   of this work is to develop a microfluidic chip called Cell Gym for   simultaneous observation of cell growth with and without Cell Exercise.   Through experiments, we observed significant difference in cell behavior with   and without Cell Exercise. Therefore, we believe it fits well to the journal   of Micromachines since it is about an application of micro-scale device for   cellular research.

Round 2

Reviewer 2 Report

The authors responded all the questions carefully and detailly. Furthermore, the whole manuscript was revised carefully according to the comments, which made the manuscript clearer and more readable. The topic of this manuscript is interesting. It seems that the microdevice used in this work could be a potential method to control the pressure onto the cells and to investigate the behavior changes of cell under different pressure exposures. So, I recommend that the manuscript could be accepted by the journal of Micromachines.

Author Response

Thanks to the reviewer for the positive comments. We really appreciate it.